# Freeze-Drying of a Capsid Virus-like Particle-Based Platform Allows Stable Storage of Vaccines at Ambient Temperature

**DOI:** 10.3390/pharmaceutics14061301

**Published:** 2022-06-18

**Authors:** Kara-Lee Aves, Christoph M. Janitzek, Cyrielle E. Fougeroux, Thor G. Theander, Adam F. Sander

**Affiliations:** 1Department of Immunology and Microbiology, University of Copenhagen, 2200 Copenhagen, Denmark; kara-lee@sund.ku.dk (K.-L.A.); chris.janitzek@gmail.com (C.M.J.); thor@sund.ku.dk (T.G.T.); 2AdaptVac, Ole Maaløes Vej 3, 2200 Copenhagen, Denmark; cfougeroux@adaptvac.com

**Keywords:** capsid virus-like particle, cVLP, Tag/Catcher, freeze-drying, lyophilization, vaccine storage, cold chain, Influenza, SARS-CoV-2

## Abstract

The requirement of an undisrupted cold chain during vaccine distribution is a major economic and logistical challenge limiting global vaccine access. Modular, nanoparticle-based platforms are expected to play an increasingly important role in the development of the next-generation vaccines. However, as with most vaccines, they are dependent on the cold chain in order to maintain stability and efficacy. Therefore, there is a pressing need to develop thermostable formulations that can be stored at ambient temperature for extended periods without the loss of vaccine efficacy. Here, we investigate the compatibility of the Tag/Catcher AP205 capsid virus-like particle (cVLP) vaccine platform with the freeze-drying process. Tag/Catcher cVLPs can be freeze-dried under diverse buffer and excipient conditions while maintaining their original biophysical properties. Additionally, we show that for two model cVLP vaccines, including a clinically tested SARS-CoV-2 vaccine, freeze-drying results in a product that once reconstituted retains the structural integrity and immunogenicity of the original material, even following storage under accelerated heat stress conditions. Furthermore, the freeze-dried SARS-CoV-2 cVLP vaccine is stable for up to 6 months at ambient temperature. Our study offers a potential solution to overcome the current limitations associated with the cold chain and may help minimize the need for low-temperature storage.

## 1. Introduction

The current SARS-CoV-2 pandemic has provided an unprecedented opportunity to evaluate different vaccine technologies and has highlighted some of the limitations in the collective vaccine supply chain and distribution capacities. Within this vaccine development landscape, modular particle-based platforms have shown remarkable promise, demonstrating high efficacy and good safety profiles. Capsid-virus-like particles (cVLPs) are self-assembling protein nanocages, which share many structural similarities with native viruses. These include their sizes [1] and repetitive surface geometries [2], which promote effective lymphatic drainage and strong B cell receptor cross-linking, leading to potent antibody responses. These, together with their excellent safety profile and good manufacturability, make cVLPs attractive platforms for vaccine development.

The Tag/Catcher AP205 cVLP platform uses a split-protein conjugation system to covalently couple vaccine antigens to the surface of pre-assembled cVLPs [3,4]. Upon mixing of the antigen and cVLP components, a spontaneous isopeptide bond is formed between the reactive Tag (a 15 amino acid peptide) and Catcher (a 12 kDa protein), which can be genetically fused to the antigen or cVLP subunit interchangeably [3]. This modular platform has been used in the development of many promising vaccine candidates [5,6,7,8,9] including a SARS-CoV-2 vaccine [10,11], which has recently shown high immunogenicity and good safety results in phase I/II clinical testing (clinical trials.gov identifiers: NCT04839146 and NCT05077267) [12,13] and which will enter a phase III clinical trial this year (clinical trials.gov identifier NCT05329220).

As with all commercially available vaccines, cVLPs require an undisrupted cold chain for transportation and distribution in order to maintain vaccine stability and efficacy. These end-to-end cold chain requirements are expensive for mass distribution [14] and make it logistically challenging to reach all target populations [15,16]. Additionally, breaches in the cold chain are relatively common and major contributors to vaccine wastage [17,18]. Hence, it is desirable to develop more thermostable vaccine formulations that can be stored at ambient temperature for extended periods, and which remain stable over broader temperature ranges.

Freeze-drying, or lyophilization, is a dehydration process in which water is removed from a product under vacuum, allowing ice to change directly from solid to vapor without passing through a liquid phase. Currently, at least 17 FDA-approved vaccines have lyophilized components, the majority of which are live-attenuated viruses [19]. However, there remain few examples of lyophilized recombinant protein/subunit vaccines. During the freezing and drying steps, proteins are exposed to several stressors, such as pH change, ice–liquid interfaces, and loss of the hydration shell [20], which collectively can compromise vaccine stability, cause aggregation, and decrease vaccine potency. However, the addition of excipients, such as sugars, polymers, and bulking agents can help mitigate these effects [21,22].

Here, we provide a proof of principle for the freeze-drying of the Tag/Catcher AP205 cVLP platform and demonstrate the long-term stability of the lyophilized material. The stabilizing effects of various buffer and excipient formulations on the unconjugated cVLP backbone were assessed. Furthermore, we evaluated the effects of freeze-drying on two model cVLP vaccines: (i) a pre-clinical influenza vaccine targeting the stem region of the hemagglutinin trimer (HA_stem_) [9] and (ii) a clinical-stage SARS-CoV-2 vaccine targeting the receptor-binding domain (RBD) of the spike protein [10]. Both vaccines maintained their stability and immunogenicity following freeze-drying when stored under both ambient and accelerated heat stress conditions.

## 2. Materials and Methods

### 2.1. Expression and Purification of Catcher-cVLP and Tag-cVLP

The components of a proprietary Tag/Catcher conjugation pair were genetically fused to the N-terminus of the bacteriophage AP205 coat protein (Gene ID: 956335) to form Catcher-cVLP and Tag-cVLP, respectively, which were expressed and purified as previously described [3]. Briefly, expression was performed in *E. coli* One Shot^®^ BL21 (DE3) cells (Invitrogen, Waltham, MA, USA), and particles were purified by ultracentrifugation over an Optiprep™ (Sigma-Aldrich, St Louis, MO, USA) density step gradient. Samples were dialyzed overnight into 20 mM of sodium phosphate (pH 7.4) or 15 mM of Tris + 200 mM sodium chloride (pH 8.5) buffer using a 1000 kDa MWCO dialysis tube (Spectrum Labs, Rancho Dominguez, CA, USA). Following dialysis, lyoprotectants (5% trehalose, 5% sucrose, 5% mannitol, or 5% sucrose + 0.005% Tween^®^20 (wt/vol)) were added and samples were diluted to a final concentration of 2 mg/mL total protein with their respective buffer.

### 2.2. Purification of HA_stem_-cVLP Vaccine

HA_stem_-cVLP was purified as described in [9]. Briefly, HA_stem_ [23] with a genetically fused C-terminal 6×histidine tag and SpyTag sequence was expressed in High-Five insect cells and purified by affinity chromatography followed by size exclusion chromatography using a Sephacryl^®^ S-300 column (Sigma-Aldrich, St Louis, MO, USA). HA_stem_.Tag trimers were mixed with purified Catcher-cVLP at a 1:1.25 (cVLP:HA_stem_.Tag) molar ratio and incubated overnight at 4 °C. Excess unbound HA_stem_.Tag was removed by density step-gradient ultracentrifugation followed by overnight dialysis into a PBS + 5% sucrose buffer (pH 7.4) using a 1000 kDa MWCO dialysis tube (Spectrum Labs, Rancho Dominguez, CA, USA).

### 2.3. Purification of RBD-cVLP Vaccine

RBD-cVLP was purified as described in [10]. Briefly, RBD (aa 319-591) with a genetically fused N-terminal Catcher sequence and C-terminal C-tag (RBD.Catcher) was expressed in S2 insect cells (ExpreS2ion Biotechnologies, Hørsholm, Denmark) and purified by affinity chromatography followed by size exclusion chromatography. Monomeric RBD.Catcher was mixed with purified Tag-cVLP at a 1:1 molar ratio in a PBS buffer supplemented with 15 mM Tris (pH 8.5) and 200 mM sucrose and incubated at room temperature (RT) overnight. Excess unconjugated RBD.Catcher was removed by dialysis against PBS + 15 mM Tris (pH 8.5) + 200 mM sucrose for 24 h using a 1000 kDa MWCO dialysis tube (Spectrum Labs, Rancho Dominguez, CA, USA).

### 2.4. Freeze-Drying and Reconstitution

Aliquots of 500 µL (Catcher-cVLP and Tag-cVLP) and 200 µL (HA_stem_-cVLP and RBD-cVLP) were rapidly frozen in liquid nitrogen and then immediately lyophilized using a Hetosicc CD52.1 freeze-dryer for 24 h at 0.1 mbar and −60 °C. No desorption step was performed. Freeze-dried (FD) samples were stored at an ambient temperature of approximately 18–22 °C (AT) or 37 °C for up to 6 months and then reconstituted to the original volume with ultra-pure water.

### 2.5. Stability Assessment

The following stability and immunogenicity studies were performed on freshly reconstituted material and compared to non-frozen samples or frozen non-freeze-dried samples that had been stored at −80 °C. The total protein concentration and percentage recovery were estimated using a BCA assay according to the manufacturer’s instructions (Thermo Fisher Scientific, Waltham, MA, USA). The relative proportion of non-aggregated particles that remained in suspension was estimated by reduced SDS-PAGE and InstantBlue Coomassie staining (Abcam, Cambridge, United Kingdom) of samples taken before or after centrifugation at 16,000× *g* for 2 min. cVLP protein bands were quantified by densitometry using Image Lab software version 6.0 (Biorad, Hercules, CA, USA). Particle integrity and RNA content were analyzed by agarose gel electrophoresis. Native samples were run on a 1% agarose gel alongside a sample that had been denatured by boiling for 5 min at 95 °C. Agarose gels were then stained with ethidium bromide and InstantBlue Coomassie. Initial Tag-Catcher conjugation dynamics were assessed in small-scale coupling reactions. Catcher-cVLP and Tag-cVLP were incubated with HA_stem_.Tag or RBD.Catcher proteins, respectively, for 1 h at RT. Samples were taken before and after centrifugation (16,000× *g* for 2 min) and run on reduced SDS-PAGE. Protein bands were quantified by densitometry, as described above, and the percentage relative conjugation was measured as the band volume of the conjugation band relative to the total lane band volume.

### 2.6. Dynamic light Scattering

For size distribution analysis, non-centrifuged samples were adjusted to 0.2–0.4 mg/mL and loaded into a disposable Eppendorf Uvette cuvette (Sigma-Aldrich, St Louis, MO, USA). Measurements were performed at 25 °C with 20 acquisitions of 5 sec each using a DynaPro NanoStar (WYATT Technology, Santa Barbara, CA, USA) equipped with a 658 nm laser. The average hydrodynamic diameter and polydispersity index (represented as percentage (%Pd)) were estimated using Dynamics software (version 7.10, WYATT Technology, Santa Barbara, CA, USA). The results shown are representative plots of measurements performed in duplicate or triplicate. For the temperature-dependent stability analysis, samples were adjusted to 0.2–0.4 mg/mL and centrifuged for 3 min at 16,000× *g* to pellet possible aggregates. A total of 20 µL of samples were loaded into a disposable microcuvette (WYATT Technology, Santa Barbara, CA, USA) and sealed with a thin layer of silicone oil to prevent evaporation. Continuous measurements (5 scans of 5 sec each) were performed between 25 °C and 80 °C with a temperature ramp rate of 1 °C/min. Onset analysis was performed using Dynamic software (Version 7.10).

### 2.7. Transmission Electron Microscopy

Samples were diluted to between 0.1 and 0.3 mg/mL and adsorbed onto fresh glow-discharged 200-mesh carbon-coated grids. Grids were washed twice with ultra-pure water and stained with 2% uranyl acetate (pH 7.0) for 1 min. Excess stain was removed by blotting with filter paper and the grids were then imaged using a CM 100 BioTWIN electron microscope (Phillips, Amsterdam, The Netherlands).

### 2.8. ACE2 Receptor Binding Assay

The measurement of RBD binding to the human ACE2 receptor was performed using an enzyme-linked immunosorbent assay (ELISA). Moreover, 96-well plates (Nunc MaxiSorp, Invitrogen, Waltham, MA, USA) were coated overnight at 4 °C with 0.1 μg/well recombinant human ACE2 protein produced in S2 cells (ExpreS2ion Biotechnologies, Hørsholm, Denmark). Plates were blocked for 1 h at RT with 0.5% skimmed milk in PBS. Soluble (unconjugated) RBD.Catcher or RBD-cVLP (stored at −80 °C for 2 months or freeze-dried and stored at ambient temperature or 37 °C for 2 months before reconstitution) was added to the plate in 2-fold serial dilutions, starting from a dilution of 100 nM. Plates were incubated for 1 h at RT before being washed 3 times with PBS and then probed with a mouse anti-RBD monoclonal antibody (produced in-house) and incubated for 1 h at RT. Plates were washed 3 times with PBS and incubated with a goat anti-mouse IgG HRP secondary antibody (A16072, Life technologies, Carlsbad, CA, USA) for 1 h at RT. After 3 washes with PBS, plates were developed with TMB X-tra substrate (Kem-En-Tec Taastrup, Denmark,) and absorbance was measured at 450 nm on a HiPo MPP-96 microplate reader (BioSan, Riga, Latvia).

### 2.9. Mouse Immunization Studies

Female BALB/c AnNRj mice (6–8 weeks old) were obtained from Janvier Labs and housed in a specific pathogen-free facility. Mice (*n* = 5–6 per group) were immunized intramuscularly with 7 µg (HA_stem_-cVLP) or 5 µg (RBD-cVLP) total protein of non-freeze-dried (stored at −80 °C for 2 months) or freeze-dried (stored at ambient temperature or 37 °C for 2 months) and reconstituted vaccine. Blood samples were collected prior to the first immunization (naive) and 2 weeks after immunization. Collected whole blood was allowed to clot overnight (at 4 °C) and serum was isolated by 2× centrifugation at 800× *g* for 8 min. 

### 2.10. Serum Immunoglobulin Levels

An ELISA was used to assess the vaccine-induced immune responses. The 96-well microtiter plates (Nunc MaxiSorp, Invitrogen, Waltham, MA, USA) were coated with 0.1 μg/well recombinant HA_stem_.Tag (produced in baculovirus expression system) or SARS-CoV-2 Spike protein aa35-1208 (produced in ExpreS2, ExpreS2ion Biotechnologies, Hørsholm, Denmark) and incubated overnight at 4 °C. Plates were blocked with 0.5% skimmed milk in PBS for 1 h at RT with shaking. Mouse sera were diluted in PBS + 0.5% skimmed milk in a 3-fold serial dilution, starting at a 1 in 50 dilution. A total of 50 μL of the diluted sera was then transferred to the plates and incubated for 1 h at RT with shaking. Plates were washed 3 times with PBS and probed with a goat anti-mouse HRP conjugated secondary antibody (anti-IgG (Life technologies, A16072); IgG1 (Invitrogen, A10551); IgG2a (Invitrogen, M32207); IgG2b (Invitrogen, M32407); or IgG3 (Thermo Fisher, M32707)) for 1 h at RT with shaking. All secondary antibodies were used at a dilution of 1:1000 in PBS + 0.5% skimmed milk. Plates were washed 3 times with PBS and developed with TMB X-tra substrate (Kem-En-Tec, Taastrup, Denmark). Absorbance at 450 nm was measured on a HiPo MPP-96 microplate reader (BioSan, Riga, Latvia).

### 2.11. Statistical Analysis

All statistical analyses were performed using GraphPad Prism (GraphPad, San Diego, San Diego, CA, USA, version 8.4.3). Statistical differences were analyzed using an unadjusted, non-parametric, two-tailed Mann–Whitney t-test, with a *p* value < 0.05 deemed statistically significant.

## 3. Results

### 3.1. Effect of Excipient Addition on the Stability of Unconjugated Tag/Catcher-cVLPs following Freeze-Drying

In order to ensure the stability of products during freeze-drying and storage, it is often necessary to include stabilizing excipients in the formulation. Sucrose and trehalose are the two of the most commonly used sugars and have well-defined cryo- and lyoprotective properties [19]. Mannitol is often used as a crystallizing bulking agent and can prevent cake collapse and improve the consistency of freeze-dried products [21]. Additionally, the combination of sugar and TWEEN^®^ 20 has previously been shown to have an additive stabilizing effect [22]. The effects of these excipients on the cake consistency and particle stability of freeze-dried Tag-cVLP and Catcher-cVLP platform backbones were assessed using sodium phosphate (pH 7.4) or Tris (pH 8.5) buffer formulations. Cake appearance can be indicative of physiochemical changes that may impact the stability and quality of the product after reconstitution. However, although cake collapse and shrinkage were observed in some of our preparations, particularly those using Tris buffer (Appendix A), they did not correlate with the degradation or instability of the particles (Appendix A). For example, the addition of mannitol led to a uniform cake solid but resulted in greater particle polydispersity after reconstitution (Appendix A). The addition of 5% trehalose or 5% sucrose + 0.005% TWEEN^®^ 20, however, resulted in particles that after reconstitution, most resembled the non-freeze-dried cVLP starting material. Thus, these formulations, together with samples lacking any stabilizing excipient were used for further longer-term stability testing. For simplicity, only the results of Catcher-cVLP formulated in a sodium phosphate buffer are shown here. Additional results for the Tag-cVLP as well as formulations in the Tris buffer are presented in the Appendix A and summarized in Appendix A. 

Catcher-cVLPs were freeze-dried and the lyophilized powder was stored at ambient temperature (FD-AT) or 37 °C (FD-37 °C) for 2 months. Following reconstitution, the stability of the freeze-dried particles was compared to a frozen, non-freeze-dried control, which had been stored at −80 °C (non-FD) (Figure 1). Transmission electron microscopy showed that Catcher-cVLPs maintained their particle integrity following freeze-drying and reconstitution, even in the absence of a stabilizing excipient (Figure 1A). Dynamic light scattering (DLS) was used to assess the average particle hydrodynamic radius and polydispersity. Rapid freezing of Catcher-cVLPs results in a small population of aggregated particles (approximately 500 nm in diameter). However, the subsequent drying step of lyophilization did not induce further aggregation but did result in a slight increase in the polydispersity (Figure 1B,E). The average hydrodynamic radius of the FD-37 °C sample lacking a stabilizer was however twice that of the non-FD and FD-AT samples (Figure 1B and Appendix A). Similarly, while all FD-AT samples and those FD-37 °C samples containing trehalose or sucrose showed a similar thermal stability profile to the non-FD control, the FD-37 °C sample lacking stabilizer had a lower onset temperature (the temperature at which unfolding/aggregation occurs) (Figure 1C). This suggests that a stabilizing excipient is required for the long-term stability of FD Catcher-cVLPs under accelerated heat stress conditions. 

During the expression of AP205 cVLPs, bacterial host RNA is encapsulated inside the particles. Agarose gel electrophoresis was used to determine whether the encapsulated RNA remains associated with the particles, or is expelled during the freeze-drying process, as well as to further assess particle integrity (Figure 1D). In order to show the relative migration of nucleic acids (stained with ethidium bromide (left panel)), relative to protein (stained with Coomassie brilliant blue (right panel)) when particles have disassembled, a Catcher-cVLP control sample was denatured by heating at 95 °C and run in parallel to the native samples. In contrast to the denatured Catcher-cVLP sample, in all native non-FD and FD samples the nucleic acid and protein components migrated to the same point in the agarose gel, indicating that the encapsulated RNA remained associated with the particles during freeze-drying. Additionally, all FD-AT samples displayed similar electrophoretic mobility compared to the native non-FD control. However, the FD-37 °C sample with no added excipient showed multiple higher-order populations, further validating the DLS stability results.

Finally, the initial conjugation dynamics of the FD Catcher-cVLP to a Tag-antigen were assessed in small-scale coupling reactions followed by SDS-PAGE analysis and densitometry. All FD samples were able to bind to the Tag-antigen and following a 1 h incubation resulted in similar Tag:Catcher conjugation levels as the non-FD control (Figure 1F). This indicates that the freeze-drying process did not affect the cVLPs ability to bind to Tagged antigens.

Comparable results were obtained for the Tag-cVLP backbone (Appendix A and Appendix A). However, Tag-cVLP appears more sensitive to both the freezing and drying steps involved in lyophilization, resulting in a more noticeable population of aggregated particles. Yet, as with Catcher-cVLP, a formulation with 5% trehalose or 5% sucrose + 0.005% TWEEN^®^ 20 helped maintain particle stability following freeze-drying and storage even under accelerated heat stress conditions. 

### 3.2. Effect of Freeze-Drying on the Structural Stability of HA_stem_-cVLP and RBD.cVLP Vaccines

HA_stem_-cVLP, an influenza vaccine targeting the conserved stem region of the hemagglutinin trimer, and RBD-cVLP, a SARS-CoV-2 vaccine targeting the receptor-binding domain of the spike protein, were used to evaluate the effect of freeze-drying on the physical and immunological properties of conjugated cVLPs. The two model vaccines were purified in antigen-specific buffers, both of which contain sucrose, and were subsequently freeze-dried. The lyophilized powder was stored at ambient temperature (FD-AT) or 37 °C (FD-37 °C) for 2 months. Following reconstitution, the stability of the freeze-dried particles was compared to a frozen, non-FD control, which had been stored at −80 °C (non-FD) (Figure 2 and Figure 3). 

Transmission electron microscopy (Figure 2A and Figure 3A) and agarose gel electrophoresis (Figure 2C and Figure 3C) of the FD vaccines indicated that particle integrity was maintained following lyophilization and reconstitution. Additionally, the RNA content in the vaccines migrated through the agarose gel together with the protein components, indicating that the encapsulated nucleic acids were not expelled from the particles under vacuum. For both vaccines, the antigen remained covalently conjugated to the cVLP and did not dissociate or degrade during the freeze-drying process or the 2 months of storage (Figure 2B and Figure 3B). Additionally, no protein loss was observed following centrifugation of the reconstituted vaccines, indicating that the cVLPs were monomodal and remained in suspension. Thus, freeze-drying did not induce particle aggregation. This was further confirmed by DLS analysis where the average hydrodynamic radius and polydispersity of the FD-AT and FD-37 °C samples were equivalent and completely overlapped with the non-FD controls (Figure 2D and Figure 3D and Appendix A). Furthermore, the thermal stability of the reconstituted vaccines was assessed by DLS over the temperature range of 25–80 °C. Freeze-dried RBD-cVLP maintained the onset temperature of the non-FD vaccine (non-FD = 48.53 °C; FD-AT = 48.75 °C; FD-37 °C = 48.47 °C) (Figure 3E and Appendix A). However, remarkably, both FD and non-FD HA_stem_-cVLPs were stable up to 80 °C; thus, the onset analysis could not be performed for this vaccine (Appendix A).

Additionally, to investigate whether freeze-drying induces deleterious conformational changes to the conjugated RBD antigen, an ACE2 receptor-binding assay was performed (Figure 3F). FD-AT and FD-37 °C displayed similar ACE2 binding kinetics to the non-FD control, and all RBD-cVLP formulations had significantly lower EC_50_ values compared to the unconjugated RBD antigen (*p* < 0.001).

### 3.3. Effect of Freeze-Drying on the Immunogenicity of HA_stem_-cVLP and RBD.cVLP Vaccines

To verify that the biological activity and immunogenicity of the lyophilized vaccines were maintained, mice were immunized intramuscularly with a single dose of reconstituted HA_stem_-cVLP or RBD-cVLP or non-FD reference samples, and serum was collected 2 weeks later. For both HA_stem_-cVLP and RBD-cVLP, no significant difference (*p* > 0.5) in the antigen-specific IgG antibody titer was detected between the control and FD-AT or FD-37 °C vaccines (Figure 2E and Figure 3G). Additionally, the FD vaccines induced similar IgG subclass profiles to the non-FD controls (Appendix A and Appendix A), indicating that lyophilization and 2-month storage at ambient temperature or 37 °C did not affect the biological response to the vaccines. 

### 3.4. Long-Term Stability of RBD.cVLP

Finally, to further evaluate the stability of the RBD.cVLP freeze-dried formulation over time, the lyophilized vaccine was stored at ambient temperature and the physical properties of the reconstituted vaccine were evaluated by DLS and SDS-PAGE analyses at the 1-, 2-, 4-, and 6-month time points (Figure 4 and Appendix A). FD RBD-cVLP maintained its average size (53 nm ± 0.6) and polydispersity (15.4% ± 0.9) compared to both non-frozen (51.3 nm ± 1.1; 11.2% ± 5.8) and non-FD controls (50.0 nm ± 0.9; 13.5% ± 0.8) and there was no evidence of aggregation or loss of vaccine integrity over the 6-month period. 

## 4. Discussion

Lack of thermostability is a major economic and logistical barrier limiting worldwide vaccine distribution. The majority of vaccines targeting infectious diseases require a continuous cold chain and refrigeration between +2 and +8 °C in order to maintain vaccine potency [24]. However, the global effort to distribute SARS-CoV-2 vaccines has added further pressure to this supply chain as two out of the first four approved vaccines require ultra-cold storage conditions (−20 and −70 °C) [25]. This requires extensive additional infrastructure, which is expensive to maintain and limits vaccine administration in remote areas. Forthcoming vaccines should therefore be compatible with formulations (e.g., via lyophilization) that allow them to be stored at ambient temperature and withstand wider temperature fluctuations, which could alleviate many of the challenges currently associated with the cold chain. We foresee that the number of modular nanoparticle-based vaccines will increase in the future; thus, it is important to analyze whether these more complex designs using platform technologies have any relative advantages or disadvantages when it comes to such a formulation.

In this study, we have investigated the compatibility of the Tag/Catcher AP205 cVLP vaccine platform with the lyophilization process. Although freeze-drying of several protein-based drug substances has been described [21,22], the physical stressors involved in freezing and sublimation are known to induce structural changes to proteins, which can expose hydrophobic residues leading to aggregation and even loss of biological activity [26,27,28]. Thus, it is important to assess whether the added complexity of modular vaccines negatively affects their tolerance to lyophilization. Here, we show that even using an elementary lab-scale machine and a non-optimized freeze-drying cycle lacking a secondary drying step, the Tag/Catcher cVLP platform is remarkably tolerant to lyophilization. In the presence of common amorphous stabilizing agents, both Tag-cVLP and Catcher-cVLP backbones maintain their structural integrity after freeze-drying and 2 months of storage at both ambient temperature and 37 °C. Likewise, freeze-drying of HA_stem_-cVLP and RBD-cVLP model vaccines did not affect their structural stability. Following 2 months of storage, the reconstituted vaccines had maintained the particle size and monodispersity of the non-FD starting material and there was no evidence of aggregation or precipitation of the antigen-cVLP complexes. For RBD-cVLP, we additionally demonstrated that the FD formulation is stable for up to six months at ambient temperature. Importantly, mice immunizations showed that the immunogenicity of the vaccines was maintained, with both FD-AT and FD-37 °C samples inducing comparable antigen-specific antibody titers to the conventionally stored formulation. 

For some antigens, even while in the liquid state, disordered intermolecular interactions between proteins conjugated to different particles can occur. This can cause cross-linking of the cVLPs resulting in precipitation of the vaccine. Thus, it could be hypothesized that conjugation of antigens onto the cVLP surface could negatively affect the tolerance of the particles to lyophilization by increasing the likelihood of partial protein unfolding inducing similar aggregation. However, this was not observed. The display of proteins on the cVLP surface seemed to have a stabilizing effect and enhanced the compatibility of the particles with freeze-drying. While unconjugated Tag-cVLP (displaying only the 15 amino acid peptide) required an excipient addition in order to maintain stability, the Catcher-cVLP tolerated freeze-drying and storage at ambient temperature, even in the absence of a stabilizing agent. Furthermore, freeze-drying and reconstitution of the fully conjugated HA_stem_-cVLP and RBD-cVLP vaccines resulted in particles that were almost indistinguishable from the non-FD samples. 

During recombinant *E.coli* expression, AP205 cVLPs encapsulate bacterial host RNA, which acts as an intrinsic adjuvant, engaging toll-like receptors (TLRs) 7 and 8 [29,30]. This drives IgG class switching [31] and is a major contributor to the immunogenicity of the cVLPs. A recent study on cowpea mosaic virus (CPMV) found that freeze-drying effectively ejected all encapsulated viral RNA from the CPMV particles [32]. This could potentially be deleterious for the development of a lyophilized cVLP vaccine, greatly diminishing its efficacy. However, here we show that freeze-drying does not remove the encapsulated RNA from AP205 cVLPs. This is evidenced by the co-migration of the RNA and protein components of the FD cVLPs through a native agarose gel as well as the maintenance of both the vaccine immunogenicity as well as the induced IgG subclass profile. 

For large-scale pharmaceutical manufacturing, freeze-drying is still considered an expensive process [33]. Accordingly, it is often only used when the high added value of the product justifies the cost or when the product’s limited thermal stability makes lyophilization essential (e.g., for live-attenuated vaccines). Therefore, optimization of the freeze-dry formulation and cycle is crucial in order to decrease the time and increase the temperature permissible for primary drying, thus reducing operating costs [34]. Therefore, the compatibility of the product with multiple buffers and excipients is a large advantage. We have shown that stable lyophilization of Tag/Catcher cVLPs can be achieved under diverse buffer conditions. This includes a pH range of 7.4 and 8.5, the presence or absence of sodium chloride, and even the use of sodium phosphate buffer salts, which has often been avoided due to early crystallization of the dibasic component inducing a drop in pH during the freezing step. Additionally, both trehalose and sucrose, the two most commonly used excipients, were able to effectively stabilize the cVLPs during freeze-drying and long-term storage, even at a relatively low concentration (5% *w*/*v*).

## 5. Conclusions

In conclusion, freeze-drying could provide a compelling opportunity for improving the thermostability of cVLP-based vaccines, which may help facilitate the rapid global distribution of these vaccines in the future. Our results demonstrate that the Tag/Catcher AP205 cVLP platform can be freeze-dried and stored under both ambient and heat stress conditions without loss of particle integrity or vaccine immunogenicity. Although freeze-dry stability might ultimately be antigen-dependent, our collective data on the stability of both unconjugated cVLP (Tag-cVLP and Catcher-cVLP) and two model vaccines demonstrate the compatibility of the modular platform technology with lyophilization under multiple buffer and excipient conditions.

## Figures and Tables

**Figure 1 pharmaceutics-14-01301-f001:**
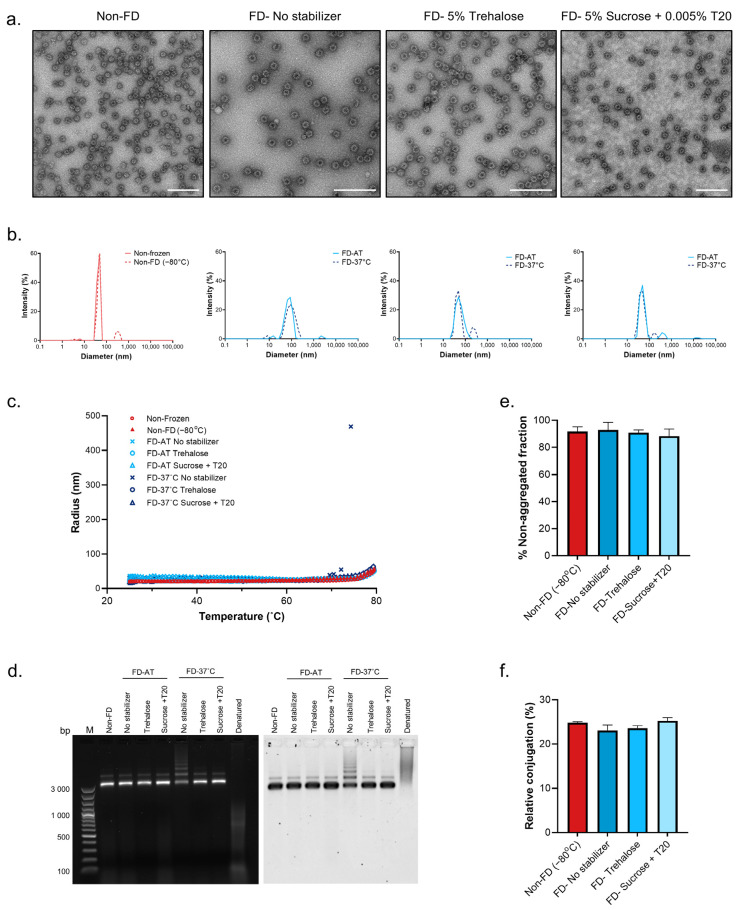
Stability of freeze-dried and reconstituted Catcher-cVLPs. Catcher-cVLPs formulated in pure sodium phosphate buffer (pH 7.4) or with the addition of stabilizing excipients (5% trehalose or 5% sucrose + 0.005% Tween^®^ 20 (T20)) were freeze-dried and subsequently stored at ambient temperature (FD-AT) or 37 °C (FD-37 °C) for 1 week (**A**) or 2 months (**B**–**F**). The stability of reconstituted freeze-dried material was compared to a non-frozen or a frozen non-freeze-dried (non-FD (−80 °C)) reference sample. (**A**) Negative stain transmission electron microscopy (TEM) images. Scale bar represents 200 nm. (**B**) Dynamic light scattering (DLS) analysis of reference samples (red) and freeze-dried Catcher-cVLP (blue) after storage at ambient temperature (solid line) or 37 °C (dashed line). (**C**) Thermal stability. The hydrodynamic radius of Catcher-cVLP was measured at increasing temperatures (from 25 °C and 80 °C) by DLS. (**D**) Agarose gel electrophoresis stained with ethidium bromide (left) and Coomassie brilliant blue (right). Native non-FD and FD samples were run in parallel to a denatured reference sample. (**E**) Quantification (by densitometric analysis of SDS-PAGE) of the relative amount (%) of reconstituted Catcher-cVLP, which remains in suspension after centrifugation (2 min at 16,000× *G*). (**F**) Quantification (by densitometric analysis of SDS-PAGE) of the relative conjugation of Catcher-cVLPs with a tagged antigen. Results show the mean and standard deviation of samples run in triplicate.

**Figure 2 pharmaceutics-14-01301-f002:**
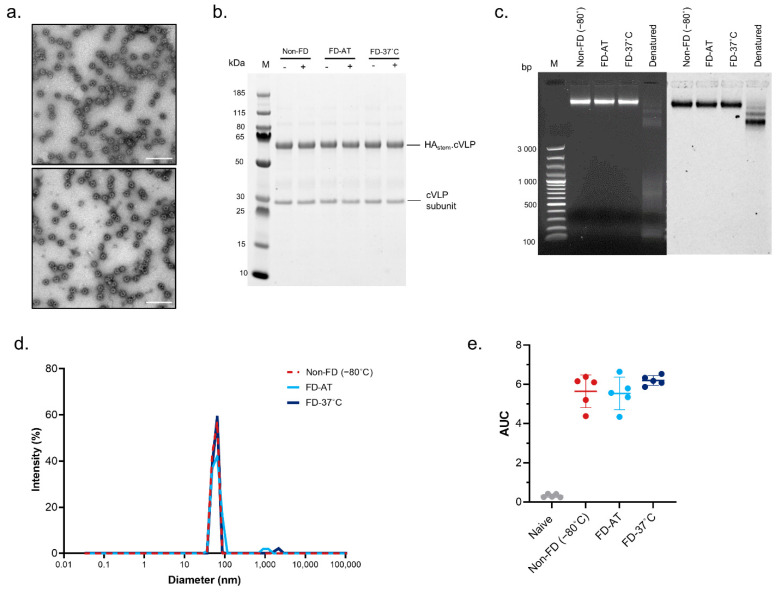
HA_stem_-cVLP influenza vaccine maintains its stability and immunogenicity after freeze-drying. HA_stem_-cVLP vaccine was freeze-dried and stored at ambient temperature (FD-AT) or 37 °C (FD-37 °C) for 1 week (**A**) or 2 months (**B**–**E**). The stability of the reconstituted material was subsequently compared to a frozen, non-freeze-dried (non-FD (−80 °C)) reference sample, which had been stored at −80 °C for the same period. (**A**) Representative transmission electron microscopy (TEM) images of non-freeze-dried HA_stem_-cVLP reference sample (top panel) and freeze-dried HA_stem_.cVLP (bottom panel). Scale bar represents 200 nm. (**B**) Representative reduced SDS-PAGE analysis of non-FD and FD HA_stem_-cVLP samples before (-) and after (+) centrifugation (2 min at 16,000× *g*). (**C**) Agarose gel electrophoresis stained with ethidium bromide (left) and Coomassie brilliant blue (right). Native non-FD and FD samples were run in parallel to a denatured reference sample. (**D**) Overlay of DLS analysis of non-FD and FD HA_stem_-cVLP. (**E**) Immunogenicity of freeze-dried HA_stem_-cVLP. Anti-HA_stem_ IgG ELISA titers measured in serum from mice (*n* = 5) immunized with either a non-FD or FD HA_stem_-cVLP vaccine. Results show the mean ± SD area under the curve (AUC) titer.

**Figure 3 pharmaceutics-14-01301-f003:**
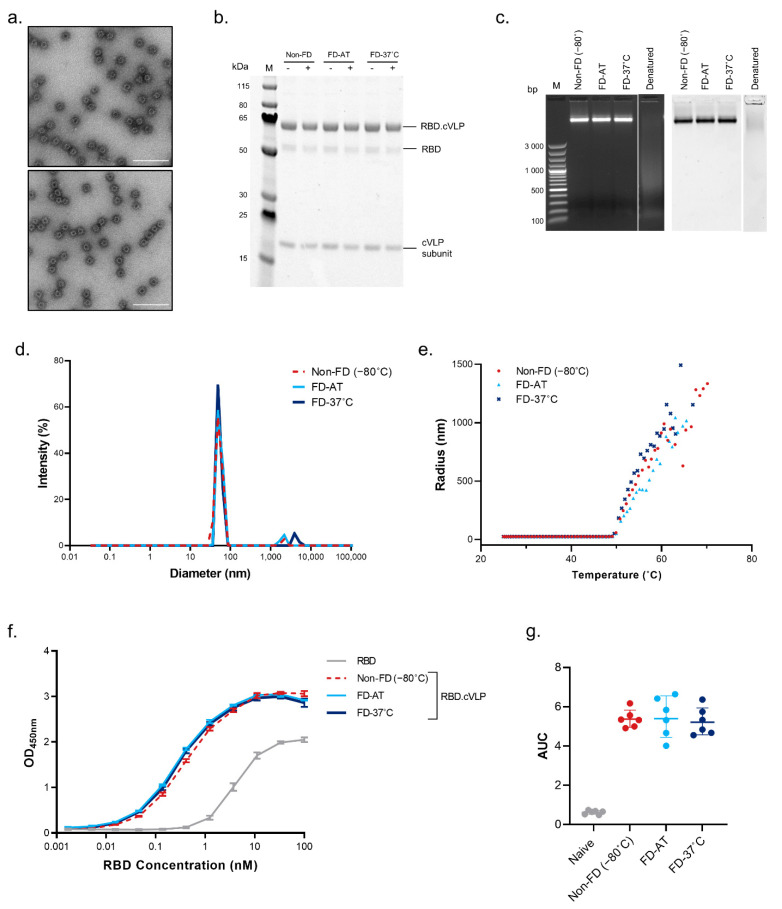
RBD-cVLP (SARS-CoV-2) vaccine maintains its stability and immunogenicity after freeze-drying. RBD-cVLP vaccine was freeze-dried and stored at ambient temperature (FD-AT) or 37 °C (FD-37 °C) for 1 week (**A**) or 2 months (**B**–**G**). The stability of freeze-dried and reconstituted material was compared to a frozen, non-freeze-dried (non-FD (−80 °C)) reference sample. (**A**) Negative stain transmission electron microscopy (TEM) images of a non-freeze-dried RBD-cVLP reference sample (top panel) and freeze-dried RBD-cVLP (bottom panel). Scale bar represents 200 nm. (**B**) Representative reduced SDS-PAGE analysis showing non-freeze-dried and freeze-dried RBD-cVLP (stored at ambient temperature or 37 °C) before (-) and after (+) centrifugation. (**C**) Agarose gel electrophoresis stained with ethidium bromide (left) and Coomassie brilliant blue (right). Native non-FD and FD samples were run in parallel to a denatured reference sample. (**D**) Overlay of DLS analysis of non-FD and FD RBD.cVLP. (**E**) Thermal stability of freeze-dried RBD.cVLP. DLS was used to measure the hydrodynamic radius of particles at increasing temperatures, from 25 °C to 80 °C. (**F**) ACE2 receptor binding assay. The binding of unconjugated (soluble) RBD and RBD.cVLP (non-FD and FD) to the human ACE2 receptor was measured in ELISA. Results show the mean ± SD absorbance of the assay run in triplicate. (**G**) Immunogenicity of freeze-dried RBD-cVLP. Anti-Spike IgG ELISA titers measured in serum samples from mice (*n* = 6) immunized with either non-FD or FD RBD.cVLP vaccine. Results show the mean ± SD area under the curve (AUC) titer.

**Figure 4 pharmaceutics-14-01301-f004:**
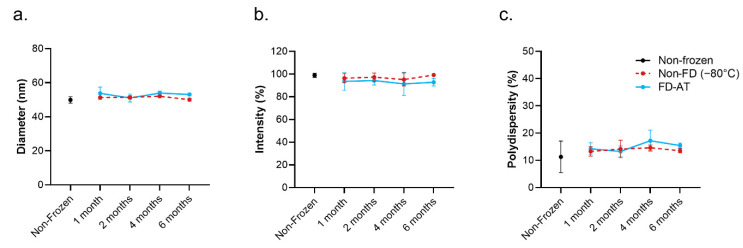
Freeze-dried RBD-cVLP (SARS-CoV-2) vaccine is stable at ambient temperature for up to 6 months. DLS measurement of (**A**) hydrodynamic diameter, (**B**) percentage intensity, and (**C**) polydispersity of the main peak population of freeze-dried RBD-cVLPs stored at ambient temperature or non-freeze-dried RBD-cVLP stored conventionally at −80 °C for 6 months compared to a non-frozen RBD-cVLP reference sample. Results show the mean ± SD of measurements performed in triplicate.

## Data Availability

Not applicable.

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
