# Peer review of "Freeze-Drying of a Capsid Virus-like Particle-Based Platform Allows Stable Storage of Vaccines at Ambient Temperature"

_pharmaceutics, 2022, doi:10.3390/pharmaceutics14061301_

Round 1

Reviewer 1 Report

Title: Freeze-drying of a capsid virus-like particle-based platform allows stable storage of vaccines at ambient temperature.

Manuscript ID: 1756371

The authors have analyzed investigate the compatibility of the Tag/Catcher AP205 capsid virus-like particle (cVLP) vaccine platform with the freeze-drying process. The authors have analyzed a combination of various excipients conditions for the biophysical properties and stability profile for the Tag/Catcher cVLPs. The authors have concluded that the freeze-dried SARS-CoV-2 cVLP vaccine is stable for up to 6 months at ambient temperatures. The authors have presented an highly relevant and impactful research through this manuscript. The authors have presented an detailed and extensive study design to prove the hypothesis of the current research. Additionally, all the sections in the manuscript are written perfectly. Although the manuscript is perfectly crafted, the reviewer would like to add the following comments to improve its quality.

Major comments:

1. The authors can add a paragraph in the results and methods section describing the formulation development, selection of excipients and a brief description about the lyophilization cycle. A graph for the lyophilization section can also be added. For example, Table 1 from supplementary figures can be added in the manuscript.

2. Stability data from Figure 1-7 in the supplementary data can be represented in tabular form and explained in a brief way in the main manuscript.

3. Similarly stability data from figure 3 and 4 from the manuscript can be represented in individual or separate tables.

4. Pictures of the lyophilized formulation showing the perfect cake can be added to the manuscript as well.

Minor comments:

1. None

Reviewer 2 Report

The work of Aves et al. entitled "Freeze-drying of a capsid virus-like particle-based platform allows stable storage of vaccine at ambient temperature " focuses on studying the compatibility of capsid virus like particle (cVLP) vaccine platform with the freeze-drying process with/without cryoprotectants. Cold chain in vaccine is one of the gran challenges in public health worldwide and lyophilization seem to be a promising alternative. I recommend that the manuscript must be accepted for publication in present form, but the authors should consider the optional amendments suggested:

-          Particle size was determined by DLS and it was expressed in terms of mean particle size and percentage in polydispersity (% Pd). First, there is a lack of the standard deviation in all the size measurements in the supporting information. Second, in DLS, the uniformity of a sample is usually expressed in term of the Polydispersity Index (PdI). The numerical value of PDI ranges from 0.0 (for a perfectly uniform sample with respect to the particle size) to 1.0 (for a highly polydisperse sample with multiple particle size populations). Values of 0.2 and below are most deemed acceptable in practice for nanoparticle materials.

In the author case, they are using the % Pd and the question is what value of %Pd is acceptable for vaccines.

-          The particle size distribution graphs could be improved by reducing the diameter scale.

-          The mean particle sizes of cVLP are around 20-30 nm according to TEM images and Table 3 (SI), but these values are not in agreement with the particle size distribution reflected in Figures 1, 2 and 3. Moreover, there is a peak size >1 µm in the particle size distributions (for example in Figure 3) which seems not to have been considered when expressing the average particle size.  To what do the authors attribute this peak? They have taken this large population into account when administering the vaccine intramuscularly?

Reviewer 3 Report

This is a very well-written report examining a crucial problem in the vaccine field. I only have minor comments:

Introduction:

1. Add references to every claim (avoid stacking references)

Discussion

1. Start with a brief summary of the main findings

2. Avoid excessive background, which should be combined with the Introduction

3. Add some limitations of the study
